# A transition to stable one-dimensional swimming enhances *E. coli* motility through narrow channels

Gaszton Vizsnyiczai [1,2,5], Giacomo Frangipane [1,3,5], Silvio Bianchi[3], Filippo Saglimbeni[3], Dario Dell'Arciprete[1,4] & Roberto Di Leonardo[1,3 ✉]

Living organisms often display adaptive strategies that allow them to move efficiently even in strong confinement. With one single degree of freedom, the angle of a rotating bundle of flagella, bacteria provide one of the simplest examples of locomotion in the living world. Here we show that a purely physical mechanism, depending on a hydrodynamic stability condition, is responsible for a confinement induced transition between two swimming states in *E. coli*. While in large channels bacteria always crash onto confining walls, when the cross section falls below a threshold, they leave the walls to move swiftly on a stable swimming trajectory along the channel axis. We investigate this phenomenon for individual cells that are guided through a sequence of micro-fabricated tunnels of decreasing cross section. Our results challenge current theoretical predictions and suggest effective design principles for micro-robots by showing that motility based on helical propellers provides a robust swimming strategy for exploring narrow spaces.

[1] Department of Physics, Sapienza University of Rome, 00185 Rome, Italy. [2] Biological Research Centre, Institute of Biophysics, Szeged 6726, Hungary. [3] NANOTEC-CNR, Institute of Nanotechnology, Soft and Living Matter Laboratory, 00185 Rome, Italy. [4] CNRS—Laboratoire de Physique de l'École Normale Supérieure, 75005 Paris, France. [5] These authors contributed equally: Gaszton Vizsnyiczai, Giacomo Frangipane. ✉email: roberto.dileonardo@uniroma1.it

Friction is usually associated to resistance against motion. However, for microorganisms friction is also the only possible source of thrust in an inertialess, low Reynolds number fluid. Of all microorganisms, bacteria have developed an amazingly simple mechanism to exploit friction for self-propulsion. It is based on a single motor unit, the flagellar motor that applies a constant torque on a thin helical filament, the flagellum. A continuously rotating flagellum allows swimming with a single degree of freedom while escaping the constraints imposed by kinematic reversibility[1]. At the same time, flagella join together in a long bundle that extends the total cell length, thus significantly reducing the angular diffusion. While still being an amazing swimming strategy in the bulk, flagellar propulsion would be useless if not robust against confinement, which is a very common feature of natural microenvironments. Confinement by solid walls leads to an increased viscous drag on the cell body, but since flagellar thrust is also derived from friction it may also increase upon confinement. Is this thrust increase enough to counterbalance the larger drag on the cell body and leave swimming speeds unaffected? By tracking individual cells, that start from the bulk and end up swimming on the surface of a flat wall, a 20% speed reduction was observed. This suggests a significant thrust enhancement which is however unable to overcompensate for the drag increase[2,3].

Very remarkably swimming speeds remain robust even when a second parallel wall is added. Biondi et al.[4] compared velocity distributions of *Escherichia coli* cells swimming in wide microchannels with heights ranging from 2 to 10 μm. Although speed variability within the same culture is much larger than the average speed variations observed in different channels, a statistically significant speed reduction was observed only for the smaller 2 μm gap, while a 10% speed increase over the bulk was reported for the 3 μm channel. This last result was not confirmed by Männik et al.[5] who used microfabricated two-dimensional chips to study the growth and motility of *E. coli* and *Bacillus subtilis* in channels with a fixed height of about 6 μm, and widths ranging from 0.3 to 5 μm. There a practically constant speed was observed until a sharp drop occurs for gaps smaller than 1.1 μm. This last findings are consistent with numerical simulations[6] where a monotonic speed decrease is predicted when one assumes that bacteria swim with a constant flagellar torque. So far, large population variability has precluded a systematic and quantitative assessment of confinement effects on bacterial motility. Furthermore, if the evidence is controversial for quasi-two-dimensional confinement, no experiments address bacterial motility in stronger quasi-one-dimensional (1D) confinement as that found in narrow blood vessels[7]. Some important questions remain open: how much can we reduce the fluid volume around a cell before substantially reducing its motility? Is there an optimal degree of confinement that can enhance motility and how?

Here we use direct laser writing to build a three-dimensional structure that guides individual cells through a sequence of microtunnels with decreasing widths ranging from 3.9 to 1.4 μm. By tracking the speed of the same cell moving through tunnels of different size, we can isolate the effects of confinement on speed from all other sources of speed variations among different cells. In contrast with theoretical predictions, we find a speed that increases as the cell moves through tunnels with decreasing cross-section. The speed reaches a maximum in tunnels of 2.3 μm width and then rapidly decreases in tighter tunnels. We demonstrate that the maximum speed occurs when swimming on the tunnel axis becomes hydrodynamically stable so that the cell swims at a higher distance from the surrounding walls minimizing cell body drag. Our results show that strong confinement provided by closely surrounding walls results in a transition to 1D swimming where bacteria only explore the neighborhood of the capillary axis

and move with a speed that is higher than in the lighter confinement conditions provided by a single flat wall.

## Results

**Speed increases with confinement.** Our structure is composed of eight square profile microtunnels with a length of 40 μm each that are connected in series by turning sections as shown in Fig. 1a. Swimming cells are guided by funneling walls towards the entrance of the largest tunnel and are then forced to pass through the eight tunnels arranged in order of decreasing width (Fig. 1b, Supplementary Movie 1). We use a non-tumbling *E. coli* strain so that cells can swim smoothly throughout the channels reducing the chance of occasional collisions and clogging of the channels. Furthermore, tumbling events in wild-type cells are not always easy to detect and may cause spurious speed fluctuations which are not related to the effects of external confinement. The bacteria used also express a red fluorescent protein that allows a clear visualization of the cell body even when closely surrounded by optically distorting walls. Discarding all bacteria that are not swimming alone in every tunnel of the sequence, we are left with 100 trajectories.

From these trajectories we isolate data points within a 25 μm range inside the tunnels starting at a distance of 10 μm from the entrance and ending 5 μm from the exit. From the clipped trajectory data we obtain the mean speed of the cell's body centroid within each microtunnel. The variability of cells' speed in each tunnel is 15% (relative standard deviation). After the cells exit the last tunnel, they are further tracked along a path of about 50 μm to extract their free speed. The mean free speed is 28.8 μm s$^{-1}$ with a standard deviation of 3.8 μm s$^{-1}$. We remark that the free speed is different from bulk speed, since outside of

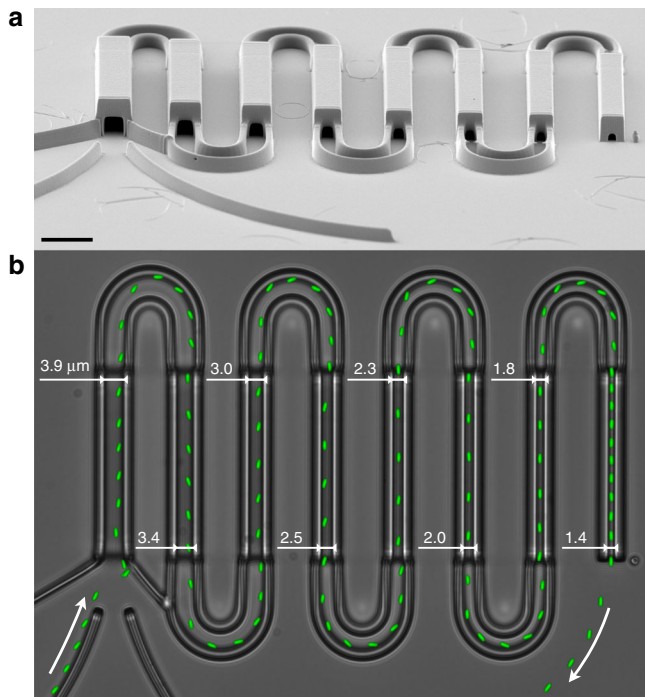

**Fig. 1 E. coli cells swimming through a sequence of microtunnels of decreasing cross-section. a** Scanning electron micrograph of the three-dimensional structure fabricated by two-photon polymerization. **b** Bacterial cells are imaged by fluorescence microscopy, a representative cell is shown in green at constant time intervals of 0.3 s. A bright-field microscopy image of the microtunnel structure is overlapped on the background. Scale bars 10 μm.

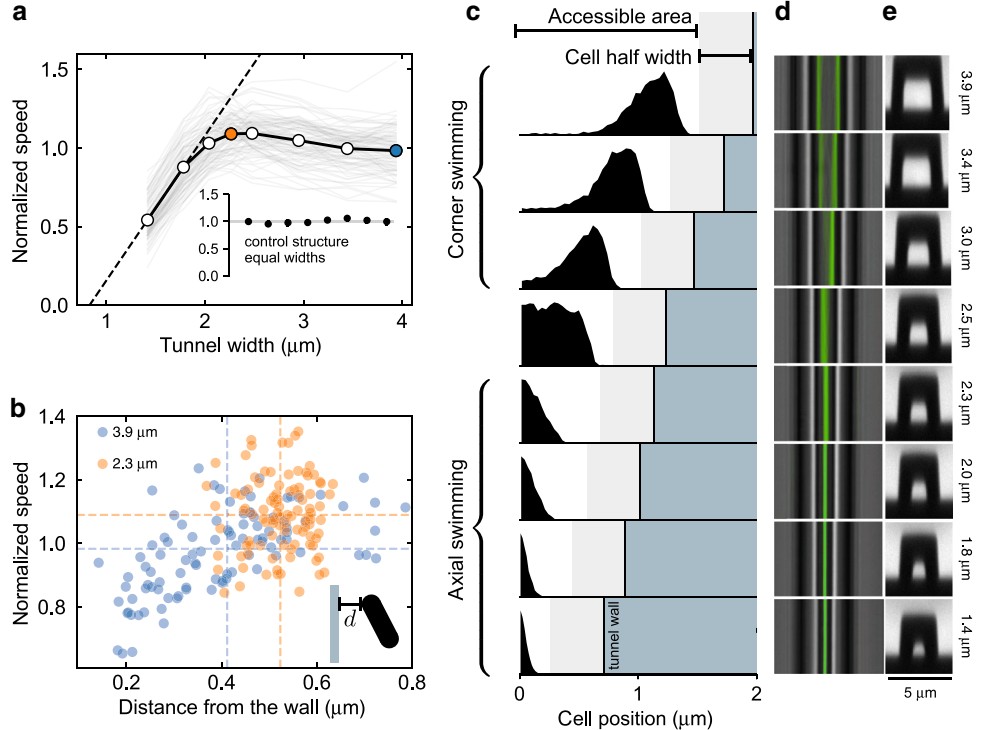

**Fig. 2 Stable swimming on the microtunnel axis enhances bacterial motility in narrow constrictions. a** Swimming speeds of *E. coli* cells as a function of microtunnel widths. Individual cell speeds are normalized to the free values measured outside of the structure and are reported as gray lines. Open circles represent speed averages on 100 cells. The dashed line is a linear extrapolation reaching zero speed for a tunnel width of 0.83 μm. The inset plots the results of a control experiment where speed is measured in a structure where all the tunnels widths are 3 μm. Symbol sizes are comparable with the standard errors. **b** Normalized speed vs mean distance from the tunnel wall for each cell in the two tunnels with widths of 3.9 μm (blue) and 2.3 μm (orange). **c** Distributions of the cell centroids positions measured from the tunnel axis. A transition to stable axial swimming is observed for tunnel widths below 2.5 μm. The maximum speed in panel **a** corresponds to the largest tunnel for which axial swimming is stable. **d** Central sections of all the tunnels with the trajectories of all the bacteria superimposed are shown. **e** Cross-sections of the tunnels obtained by two-photon microscopy (see Supplementary Note 1).

the structure cells are trapped by the glass slide surface[2,3,8–10]. However, these surface bound bacteria typically represent the majority of cells found in mild confinement conditions.

Swimming speeds versus tunnel widths are shown for all bacteria as gray lines in Fig. 2a, where, for every cell, we normalize speeds to the corresponding free value. Open circles in Fig. 2a represent mean values averaged over all bacteria. These data show that: (i) in the widest channel the average speed is the same as the free speed; (ii) the average speed increases in tighter channels until it reaches a maximum value of 1.1 times the free speed for a channel width of about 2.3 μm; (iii) for smaller tunnel widths the speed then rapidly decreases, although cells can still swim at 50% of the free value in the smallest tunnel having a width of only 1.4 μm. Interestingly, if we extrapolate the last two points of the averaged speed curve, we find that the speed vanishes for a tunnel width of 0.8 μm, corresponding to the average cell body thickness. Despite the large speed differences among cells, the standard error of normalized mean speeds in Fig. 2a is always within the symbols size. The observed initial speed increase is statistically significant with a *p* value for the comparison between tunnels 2.3 μm (orange) and 3.9 μm (blue) given by $p = 10^{-8}$ (Mann–Whitney). Furthermore as a control experiment we fabricate a second structure composed of equal sized tunnels (width 3 μm) to check for the presence of any intrinsic cellular factor that could give rise to a systematic speed variation as the cell traverses the entire structure. Results are reported in the inset of Fig. 2a and show no statistically significant differences between-group means as determined by one-way

ANOVA ($p = 0.95$). One might think that oxygen depletion can affect the swimming speed when the cell is in a narrow tunnel. However, a simple estimate already shows that this effect is negligible. Typical oxygen consumption rates in *E. coli* is $Q \approx 20$ amol min$^{-1}$ cell$^{-1}$ (ref. [11]). Assuming steady-state diffusion, this consumption rate will be equal to the oxygen flow through the tunnel cross-section. The resulting relative concentration variations will be then $\Delta C/C \approx Ql/(CDs^2) \approx 10^{-3}$ where we used $D \approx 2 \times 10^{-5}$ cm$^2$ s$^{-1}$ for oxygen diffusion coefficient in water[12], $C = 1$ amol μm$^{-3}$ for oxygen concentration in water[13] while $l$ and $s^2$ are typical length and cross-sectional area of our tunnels.

Numerical solutions[14] show that the speed of a force-free helix, driven by a constant torque in a circular microcapillary, weakly grows with confinement until the gap between the capillary walls and the flagellum reaches a value that is about 0.5 μm. However, numerical simulations using finite element methods show that, when a full cell is considered, the cell body, which is thicker than the bundle, will feel earlier the presence of the wall and counteract the speed increase predicted for the isolated helix with a rapidly increasing drag coefficient. As a result, the speed of a full cell is predicted to decrease monotonically to 90% of the bulk value for $\approx 3$ μm diameter and then quickly drop to zero[15]. This is in strong contrast with our experimental observation of a speed that initially increases with confinement. This apparent contradiction can be resolved by looking at correlations between the cell's speed and spatial arrangement inside the tunnels. In Fig. 2b we report a scatter plot of the cell's normalized speeds and their distance $d$ from the closest wall. The data of the largest tunnel are shown as

blue circles while orange circles refer to the 2.3 μm tunnel where the speed is maximal. In the larger tunnel we find a strong correlation ($r = 0.56$, $p = 10^{-9}$) showing that the cells that swim slower than their free speed are also significantly closer to walls. Surprisingly, cells in the tighter tunnel never seem to come too close to the wall which results in a higher average speed. This suggests a picture where the observed speed increase with confinement is actually due to a counter-intuitive effect where increasing confinement leads to a progressive increase of the distance from the wall.

**Axial swimming becomes stable in small tunnels**. To assess this idea more precisely, we plot for every channel the histogram of the instantaneous absolute distance of all the cells from the tunnel axis (Fig. 2c). The evolution of histogram shape with increasing confinement reveals the existence of a critical tunnel width marking a transition to 1D swimming where cell positions become narrowly distributed around the tunnel axis. In particular, 1D swimming appears to be stable in all tunnels having a width smaller than or equal to 2.3 μm. In this last tunnel, the largest for which 1D swimming is stable, we also find the maximum value of the swimming speed. The speed gain in tight channels is then associated to a transition to a stable axial swimming configuration that keeps the cell farther away from the walls. In this scenario we would then expect to find a higher speed gain in those cells that swim closer to walls in large tunnels. Figure 3a (bottom) shows that the distance from the wall $d$ in the largest tunnel (3.9 μm) is controlled by swimming characteristics such as wobbling amplitude $w$ and the pitch angle $\theta$ (see Supplementary Note 4 for more plots). In particular, cells that wobble less or swim with a higher pitch angle tend to maintain a smaller distance from the wall. Those cells are also the ones undergoing the largest displacement away from the walls when 1D swimming along the tunnel axis becomes stable. This results in a higher speed gain with confinement, as shown in the top panels of Fig. 3a. If we now select those cells that are simultaneously below the 20th percentile in $w$ and above the 80th percentile in $\theta$ and plot their average speed versus tunnel width, we obtain a curve showing a marked speed gain reaching a 40% increase over the speed in the largest channel (purple line in Fig. 3b, top panel).

Conversely, choosing those cells that are simultaneously below the 20th percentile in $\theta$ and above the 80th percentile in $w$, no significant speed increase is observed as the cells move through progressively smaller tunnels (green line in Fig. 3b, top panel). As expected, this different behavior of the speeds in the two groups is linked to a qualitatively different evolution of the wall distance with increasing confinement (Fig. 3b, bottom panel). While cells that swim closer to the walls in the entrance tunnel display a marked jump to a larger distance when 1D swimming becomes stable, those cells that keep a larger distance from the wall in the largest channel smoothly move closer to walls as the size of the channel is reduced.

In order to locate more precisely the threshold tunnel size for stable axial swimming we introduce a minimal model to describe position fluctuations along a transverse coordinate $x$. For axial swimming to be stable, a restoring speed component should appear on the cell center as it moves away from the axis ($x = 0$). For small displacements we may assume that this speed component is linear in $x$:

$$\dot{x} = v_x(x) + \eta \approx -\alpha x + \eta, \qquad (1)$$

where $\alpha$ is the slope $\partial v_x / \partial x$ and $\eta$ represents an effective white noise term $\langle \eta(t)\eta(0)\rangle = A\delta(t)$ incorporating random speed fluctuations due to both thermal and active fluctuations in flagellar dynamics. A positive $\alpha$ means stable axial swimming while a negative $\alpha$ leads to unstable axial swimming. In this situation we would expect to describe the $x \sim 0$ part of the histograms in Fig. 2c with a Gaussian distribution $P(x) \propto \exp[-\alpha x^2 / A]$. Figure 4a confirms that the logarithm of the distribution of cell positions is well fitted by a quadratic law where the coefficient of the quadratic term can be assumed to be an indicator of the slope $\alpha$. If we now plot $\alpha$ as a function of tunnel width we find a nice linear curve crossing zero for a channel size of 2.4 μm (Fig. 4b). We evaluate $\alpha$ also numerically using a Rotne–Prager method (see Methods) to simulate a full cell swimming at different lateral displacements from the tunnel axis. Numerical values for $\alpha$ are reported on top of Fig. 4b. Although experimental and numerical values for $\alpha$ can be only compared within an unknown multiplication factor, we find a remarkable agreement with experimental data with an $\alpha$ value that goes to

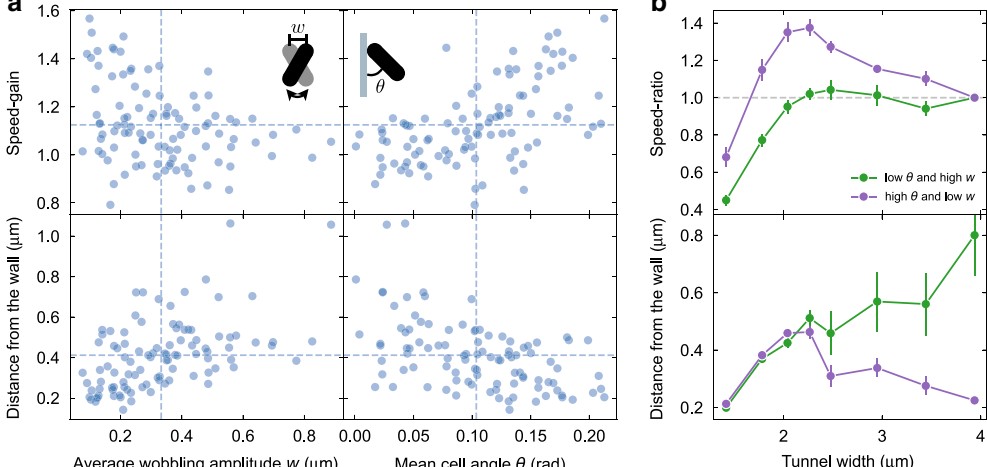

**Fig. 3 Speed gain depends on swimming characteristics. a** Speed gain (from 3.9 to 2.3 μm) and wall distances versus wobbling amplitude $w$ and pitch angle $\theta$. Each cell appears as a point in the scatter plots. Dashed lines represent mean values. Wall distance, $w$, and $\theta$ are all measured in the largest tunnel (3.9 μm). Error bars (s.e.m.) are always within the symbol size. **b** Cells with low $w$ and high $\theta$ display a substantial speed increase with confinement (purple) which is connected to a marked increase in the distance from the wall below 2.5 μm. The speeds are normalized to the values in the largest microtunnel. No significant speed gain is observed for cells with high $w$ and low $\theta$ (green) for which the wall distance progressively decreases with confinement. Plotted error bars represent s.e.m.

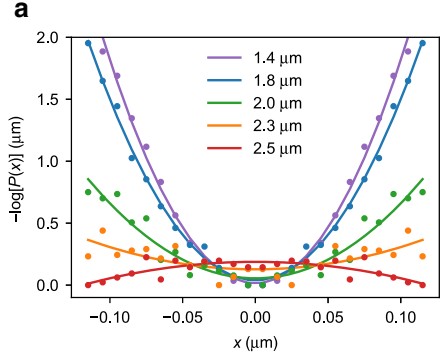
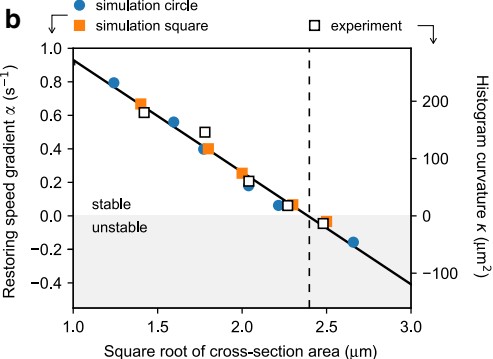

**Fig. 4 Stability of the axial swimming state. a** Effective potential for the cell center along the transverse coordinate in the five smallest tunnels. The potential is computed as the logarithm of the cell position probability distributions shown in Fig. 2c and shows a minimum where axial swimming is stable. Each solid line plots the best quadratic fit to experimental data. **b** Comparison between the curvatures of the fitted effective potential $\kappa$ (white squares) and the restoring speed gradient $\alpha$ (see Eq. (1)) obtained from numerical simulations. Orange squares and blue circles plot simulations results for tunnels with square and circular section respectively.

zero for a tunnel width that is very close to the experimental value. In simulations we can easily check for the robustness of this result for different shapes of the tunnel cross-section. Repeating the calculations for a circular tunnel we find that $\alpha$ values can be scaled on top of each other by expressing tunnel widths as the square root of cross-sectional area (side length for square tunnels and $\sqrt{\pi}\times$ radius for circular tunnels)

**Confinement enhances flagellar thrust**. In the tightest tunnel we observe an average swimming speed $U'$ that is reduced to 50% of the free value $U^0$. This is a rather modest reduction when one considers that, in the same tunnel, a spherical particle with a diameter equal to the cell body width would experience a ten-fold drag increase compared to the bulk[16]. This suggests that such a strong drag increase is probably compensated by an increase of the thrust due to confinement. Such an increase in the thrust was predicted theoretically[14] but never measured directly since most of the experiments look at the cell speed which depends on both thrust and drag. Before we begin a quantitative discussion on the thrust, some definitions are needed. Linearity of Stokes equations links forces and velocities in a rigid body through a constant resistance matrix. Treating cell body and flagellar bundle as two hydrodynamically uncoupled rigid bodies that are rigidly connected but can rotate independently around a common axis we can write

$$\begin{pmatrix} F \\ T \end{pmatrix} = -\begin{pmatrix} A & 0 \\ 0 & B \end{pmatrix} \cdot \begin{pmatrix} U \\ \Omega \end{pmatrix}, \qquad (2)$$

$$\begin{pmatrix} f \\ \tau \end{pmatrix} = -\begin{pmatrix} a & c \\ c & b \end{pmatrix} \cdot \begin{pmatrix} U \\ \omega \end{pmatrix}, \qquad (3)$$

where $F, f$ and $T, \tau$ are the axial components of the viscous forces and torques acting, respectively, on the cell body and the flagellar bundle. $U$ is the cell axial speed, and $\Omega$ and $\omega$, respectively, are the body and bundle angular velocities. Matrix elements are $A, a$ translational drag of body and bundle, $B, b$ rotational drag of body and bundle, $c$, the helical bundle coupling coefficient[16]. We define the thrust as the force transmitted by the rotating bundle to the cell body when the cell body is kept fixed by an external force ($U = 0$). From Eq. (3), this force is $f_t = -c\tau/b$. This definition of thrust has the advantage of being a property of the bundle alone with no reference to the actual drag on the load (cell body). By imposing force free ($F + f = 0$) and torque free ($T + \tau = 0$) conditions we can solve for the cell speed:

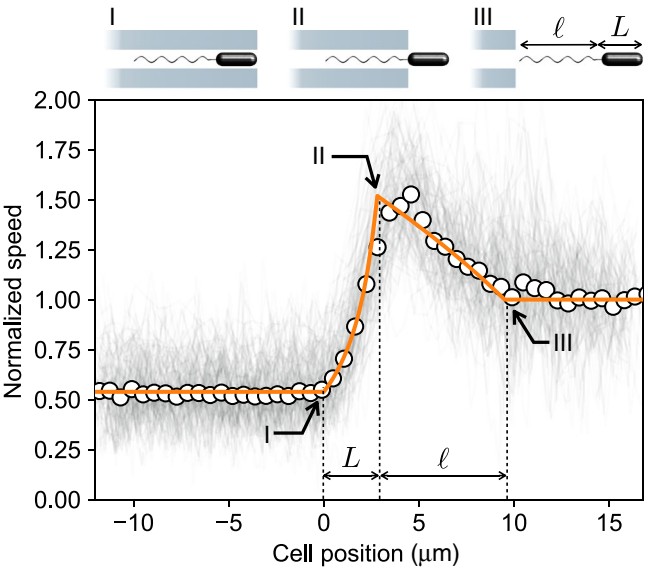

**Fig. 5 Confinement enhances flagellar thrust.** Gray lines plot the time evolution of the speed as the cell exits the tightest microtunnel at the end of the structure. Circles represent mean values averaged over all the cells. Orange solid line is the best fit with the model described in the text.

$$U = -\frac{c\tau}{b(a+A)-c^2} \approx -\frac{c\tau}{b(a+A)} = \frac{f_t}{a+A}. \qquad (4)$$

A direct evidence of the fact that confinement results in an increased thrust is obtained by plotting the speed of the cells while they progressively move across the last tunnel exit (Fig. 5). Before point I the entire cell swims inside the tunnel with an average normalized speed equal to 0.5. Between point I and II, the cell body passes through the tunnel exit, its drag is progressively reduced to the free value while flagellar drag and thrust remain constant. This results in a three-fold increase in the speed. From II to III the bundle goes from being entirely in to entirely out. Calling $a^0, A^0, f_t^0$ the drag coefficients and the thrust of a free cell and $a', A', f_t'$ the corresponding values when fully confined inside the tunnel, the thrust enhancement ratio is $f_t'/f_t^0 = (U_{II}/U_{III})(a'+A^0)/(a^0+A^0)$. Since $a' > a^0$ and, from data in Fig. 5, $U_{II} > U_{III}$ we can unequivocally conclude that $f_t' > f_t^0$. We

now propose a theoretical minimal model for the speed versus position curve in Fig. 5. Assuming that flagellar motors apply a constant torque[17,18] (see Supplementary Note 3) the speed $U$ will depend on cell position only through the drag coefficients $A$, $a$, $b$, $c$. As before we call $A^0$ and $A'$ the free and confined values of the translational drag on the cell body. We then assume that when a fraction $x/L$ of the body is outside of the tunnel, the drag will be given by the weighted average $A(x) = A' + (x/L)(A^0 - A')$ where $L$ is the body length and $x$ is the position of cell body head relative to the microtunnel exit point. Similarly the linear drag coefficient of a partially confined flagellar bundle is $a(x) = a' + ((x - L)/\ell)(a^0 - a')$, where $x - L$ is the position of the bundle end attached to the cell and $\ell$ is the bundle length which we leave as a free fitting parameter. In the same way we obtain expressions for $b(x)$ and $c(x)$ and substitute all position dependent coefficients in Eq. (4). The parameters $A^0, a^0, b^0, A', a', b'$ are obtained from numerical calculations (see Methods) over a range of $\ell$ values while $c^0$ and $c'$ are left as free fitting parameters. The best-fit curve is plotted as a solid line in Fig. 5 and provides a very good representation of data points. We find a value for $\ell = 6.7 \pm 0.3$ µm that is compatible with literature data[19]. While the translational drag on the cell body increases by a substantial factor $A'/A^0 = 3.85$, both translational and rotational drag on the flagellar bundle are less affected and increase respectively by $a'/a^0 = 1.54$, $b'/b^0 = 1.26$. Remarkably, the coupling coefficient, that is responsible for thrust generation, increases by about a factor of two $c'/c^0 = 1.95$ with a corresponding thrust enhancement of $f'_t/f^0_t = (c'/b')/(c^0/b^0) = 1.56$.

At this point it comes natural to wonder if swimming efficiency is also enhanced by confinement. There are mainly two ways of defining the self-propulsion efficiency of a swimmer. The first one is by the ratio of swimming speed $U$ over the bundle rotational frequency $\omega/2\pi$. This ratio has the dimensions of a length and represents the distance traveled by the cell in a full rotation period of the bundle. Assuming a constant torque, the energy supplied by a flagellar motor in a full rotation cycle is constant so that this definition corresponds to what we generally use for transport vehicles, i.e. the ratio of distance traveled per unit of fuel consumed. With a little bit of manipulation this efficiency can be expressed as $\epsilon_1 = 2\pi U/\omega = 2\pi bU/\tau \propto bU$ where we have used $\tau \approx b\omega$. Alternatively, swimming efficiency is often defined as the ratio between the power required to drag a dead cell body at a speed $U$ and the power $\tau\omega$ supplied by the flagellar motors to self-propel the cell at the same speed $\epsilon_2 = (A + a)U^2/\tau\omega = cU/\tau \propto cU$ where we have used the approximation $U \approx c\tau/b(A + a)$ in Eq. (4). It is interesting to note that both efficiencies are proportional to the product of cell speed by a drag coefficient in the bundle resistance matrix. If the drag coefficients increase under confinement, it can be concluded that a higher swimming speed also corresponds to a higher swimming efficiency. The situation is not so straightforward in the last two channels where the speed falls below one but the corresponding efficiencies could still be greater than outside if the coefficients $c$ and $b$ are sufficiently larger than in free cells. In the case of the last tunnel we have reliable estimates for the ratios $c'/c^0$ and $b'/b^0$ obtained from the fit of data in Fig. 5. Substituting those values we can obtain the efficiency ratio between the last tunnel and the free case. We find that the first efficiency decreases in the tightest tunnel $\epsilon'_1/\epsilon^0_1 = (b'/b^0)(U'/U^0) = 0.68$. The second efficiency is remarkably still the same as outside $\epsilon'_2/\epsilon^0_2 = (c'/c^0)(U'/U^0) = 1.95 \times 0.5 = 1$, indicating that flagella shape seems to be optimized for maximum efficiency even under strong confinement conditions[14]. The possible existence of stable swimming states at the midplane between two parallel flat walls was suggested by some theoretical studies[20,21]. However the only, though indirect, evidence for that was obtained for artificially elongated cells that swam in straight lines between nearby flat walls rather then tracing circular paths that are typical of surface swimming[22]. Theoretical work on squirmers in a capillary tube suggests instead that, for swimmers of the pusher type like E. coli, axial swimming is always unstable[23]. Here we provide the first direct evidence that E. coli cells can float away from boundaries and swiftly move at the center of a microchannel having a cross-section below a threshold size. The existence of this hydrodynamically stable state, coupled with a substantial increase in the thrust force generated by the bundle, results in a motility that is enhanced in narrow channels compared to weaker confinement conditions. This phenomenon could be relevant for other microswimmers such as sperm cells[24], blood vessels pathogens, or microrobots that are designed to explore narrow spaces[25]. Our results also challenge current theoretical and simulation schemes to identify the essential conditions for stable axial swimming and to provide a quantitative match with the observed values for the thrust enhancement.

## Methods

**Microfabrication.** Microfabrication is carried out by a custom built two-photon polymerization setup, described previously[26]. The microtunnel structures are created from SU-8 2015 photoresist (MicroChem Corp). A high numerical aperture oil immersion objective (Nikon Plan Apo Lambda ×60 1.4) is used to create a single fabrication focus with 2.3 mW optical power that is moved with 100 µm s⁻¹ scanning speed during fabrication. After exposure, the SU-8 photoresist sample is baked at 100 °C for 7 min, then developed by its standard developer solvent, and finally rinsed in a 1:1 mixture of water and ethanol. Strong adhesion of the SU-8 structures to the carrier coverglass is ensured by a layer of OmniCoat adhesion promoter (MicroChem Corp). The precise sizing of the microtunnels is ensured by measuring their widths and heights with two-photon fluorescence laser scanning microscopy (Supplementary Note 1), giving mean widths of [1.42, 1.78, 2.04, 2.27, 2.48, 2.95, 3.44, 3.94] µm.

**Microscopy and cell tracking.** Epifluorescence imaging is performed on an inverted optical microscope (Nikon TE-2000U) equipped with a ×60 (NA = 1.27) water immersion objective and a high-sensitivity CMOS camera (Hamamatsu Orca Flash 4.0). A high power LED (Thorlabs M565L3) provides the epifluorescence excitation light. Image acquisition during the experiment is run at 50 frames per second. The trajectories of the recorded cells are calculated with a custom made OpenCV based tracking software written in Python. The position of the wall appearing in Figs. 2c and 3b can be accurately determined by aligning the cells trajectories in a coordinate system where the center of the smallest tunnel is located at $x = 0$. Small drifts of the sample are corrected by setting to zero the mean position of each cell in the smallest tunnel. The wall positions in this coordinate system are then calculated by considering the spacing (15 µm) of the microtunnels and their widths which are extracted from scanning electron microscope images and two-photon microscopy as described in Supplementary Note 1 and reported in Fig. 2d.

**Cell growth and sample preparation.** For these experiments we use the smooth swimming E. coli strain HCB437[27]. The cells are modified to express the red fluorescent protein mRFP1 under the control of the lacI promoter (BioBricks, BBa_J04450 coding device inserted in pSB1C3 plasmid backbone). Single colonies of the cells are inoculated in 10 ml of LB medium, before growing overnight at 33 °C. The saturated culture is then diluted 1:100 (50 µl in 5 ml) into tryptone broth fresh medium and grown up to OD590 ≈ 0.8 at 33 °C shaken (for aeration) at 200 r.p.m. The production of mRFP1 is induced during the last growth stage by addition of 1 mM IPTG. In all culturing stages 25 µg ml⁻¹ kanamycin and 34 µg ml⁻¹ chloramphenicol are present. Bacterial cells are then harvested from culture media by centrifugation at $1100g$ for 5 min at room temperature. The pellet is re-suspended by gently mixing in motility buffer [10 mM potassium phosphate (pH 7.0), 0.1 mM EDTA (pH 7.0), and 0.02% Tween 20][28]. To increase the speed of the cells, glucose 10 mM is added to the motility buffer[11]. The cells are washed three times to replace growth medium with motility buffer. Motility buffer sustains bacterial motility but not growth/replication so that the bacterial population remains constant during the experiment. The experiment is performed in an open chamber sample built around a microtunnel structure fabricated onto a microscope coverglass. The sample is first filled with motility buffer, and then a low-density suspension of E. coli cells is added.

**Hydrodynamic simulation.** We have used the Rotne–Prager method[29,30] to numerically calculate the hydrodynamic drag coefficients of an E. coli cell body and flagella. The Rotne–Prager mobility matrix is computed by a CUDA kernel. The cell body and the flagella are simulated in two different geometries: (1) over a flat surface at a distance of 0.9 µm, as measured in the largest tunnel (Fig. 2c) and (2) on the axis of a 40 µm long square-section microtunnel. The cell body is

constructed as a spherocylinder with 2.8 μm length and 0.9 μm diameter built up from 50 nm radius spheres. The bundle is composed of 30 nm radius spheres forming a 6.9 μm long helix with a wavelength of 2.3 μm and a radius of 0.3 μm[19]. The microtunnel is constructed by 138 nm radius spheres, while the flat surface by 100 nm radius spheres. For representative plots of the model geometries see Supplementary Note 2. Our simulation calculates a bulk drag of $1.40 \times 10^{-2}$ pN s m$^{-1}$ for a cell body, close to the experimentally measured value of $1.48 \times 10^{-2}$ pN s m$^{-1}$ reported in literature[31].

To compute the restoring speed gradient $\alpha$ in Eq. (1) we simulate a full cell where two equal and opposite torques are applied on the cell body and on the flagellar bundle, respectively. While the cell orientation is kept parallel to the channel, the cell is displaced laterally by a distance $x$ from the channel axis and the transversal speed component $v_x$ is computed. For small displacements, $v_x$ grows linearly with $x$ with a slope $\alpha$ (see Supplementary Note 2).

## Data availability
The data that support the findings of this study are available from the corresponding author on reasonable request.

## Code availability
The code is available from the corresponding author upon reasonable request.

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

## Acknowledgements
This project has received funding from the European Research Council (ERC) under the European Union's Horizon 2020 research and innovation programme (grant agreement no. 834615). We thank Lóránd Kelemen (Institute of Biophysics, Biological Research Centre, Hungary) and the CNIS Labs of Sapienza University for allowing access to their scanning electron microscope.

## Author contributions
G.V., G.F., and R.D.L. designed experiments. G.V., G.F., F.S., and D.D.A. performed experiments. G.V. and S.B. designed and fabricated microstructures, and carried out the size measurement of microtunnels. G.F. carried out the tracking of the recorded cells. G.F., F.S., and D.D.A. were responsible for the growth of bacterial strains. G.V., G.F., and R.D.L. analyzed the data. G.V. and S.B. performed the hydrodynamic simulations. G.V., G.F., S.B., and R.D.L. wrote the manuscript.

## Competing interests
The authors declare no competing interests.
