## [Peer Review File · Nature Communications]

Reviewers' comments:

Reviewer #1 (Remarks to the Author):

In this work, Vizsnyiczai, Frangipane, Di Leonardo et al. present the use of 3D printed microchannels of decreasing size to study the effect of constriction on the movement of a non-tumbling *E. coli* strain. The authors show that a channel size of 2.3 μm is optimal for high-speed navigation and the dependency of the swimming speed is correlated to the position of the cell within the channel. The experimental set-up and the subject is of high relevance for the microrobotic community. However, there are several points that require clarification before publication can be granted. Below, you find a list of such points:

- The title is very general and points towards bacterial motility and narrow channels. Here, only a very limited set of conditions is tested. This reduced set clearly limits the validity of the work actually done and/or calls for a change of the title. This also points towards some questions regarding the claim of the authors from a biological point of view in the absence of proper controls:
 - o The authors only use a mutant *E. coli* strain. I fully understand the practical aspects associated with this type of bacteria, however, do the wild-type *E. coli* or the K12 mutant and other species show the same behavior?
 - o Along the same line, working with another species would give information on the parameters used for this type of cell. Is the 2.3 μm a general feature or it is related to the cell size and / or to the flagellar apparatus dimension? What about the channel geometry? Would another morphology (circle cut) have a similar outcome? These questions have been ignored and should be commented if not directly tackled.
 - o The authors only study the paths of the bacteria towards smaller channels. Even for microorganisms, a "fatigue" can be observed. Therefore, the authors have to analyze the track of bacteria starting from smaller channels towards larger channels and check if similar results are obtained.
 - o On the same line, the authors analyze the speed of the cells while exiting the channel, they have to provide data for the bacteria entering the channel too and compare the entry speed with the exit speed.
- What is the statistical relevance of this work? This may be more a biologist question as it might be clear for the physicists but the authors limit their work to 100 tracks. Again, I understand it may be difficult to track the bacteria in these conditions, but due to the fact that the authors based all their claims only working on average value of highly diverging data, they should provide a statistical analysis attesting the relevance of the claims. Indeed, the average normalized speed difference reported in the channels varies from about 0.9 to 1.1 for channel sizes varying from 2 to 4 μm whereas it seems the actual normalized speed spans from 0.6 to 1.4.

Looking forward to discussing these issues with the authors

Damien Faivre

Reviewer #2 (Remarks to the Author):

The submitted manuscript is focused on the hydrodynamics of bacterial swimming in relatively narrow channels of different width. The authors report the increase in the swimming velocity of an individual bacterium if the channel width is around 2-3 micron. This increase is associated with the transition to stable swimming straight along the center line of the channel. I believe that the paper provides a new and important experimental input for understanding the bacteria hydrodynamics in microchannels and may be considered for publication in Nature Communications after the author clarified several points.

Main points to address.

The results on bacterial swimming velocity vs channel width are presented in a clear and well-organized manner. The authors emphasize some apparent contradictions with previous studies [14] and resolve it by pointing to a difference in positions of the bacterium inside the channel. While this is correct, I believe that there is an opportunity to increase the depth of analysis by presenting additional data. It would be nice to track the flagella rotation rate vs channel width. It is interesting to see how the frequency changes with the width. If the frequency does not peak at ~ 2.2 μm width then the bacterium is swimming more "efficiently", traveling farther per revolution of flagella due to change in the shape of microscopic hydrodynamic flows. The rotation rate for the same bacterium is at a nearly constant ratio with the bacteria wobbling frequency. Since the amplitude of wobbling was tracked I assume that this data on frequency should be relatively easy to extract from already collected experimental observations.

Alternatively, the author may perform another series of experiments with fluorescently marked flagella. It will provide more accurate and detailed data.

On the downside, I found the discussion of the thrust increase in a narrow channel by analysis of speed variation near the channel exist a little weak and speculative. First of all, I suggest to define the thrust as "the force that is required to keep the cell at rest " before the analysis, not after. Otherwise, the thrust may be assumed to be equal to the drag in magnitude and opposite in direction. Then it is not clear why it's not changing between points I and II. Second, the author did not provide expressions for $a(x)$, $b(x)$ and $c(x)$ calling them "similar". That confused me. I can assume these expressions should include flagella length as a parameter, which should also be fitted. Finally, the assumption that the torque applied to flagella is constant is weak. The torque is indeed nearly a constant (with some limitations) under a relatively heavy load. A freely swimming bacterium does not utilize the maximum torque, therefore it is not constant under any load. (Moreover, it may have some memory effect and does not change instantaneously but it's the next degree of complexity). I would like to hear the authors comment on that.

Other questions and suggestions:

1. The vertical profile of the channel is not rectangular but has rounded corners. What is a typical radius? When the size of a channel is decreasing, this may become important and affect the localization of a bacterium inside the channel. How important is the transition from rectangular to circular confinement?
2. How do the authors define the position of the wall? Was it done by processing the images similar to Figure2d?
3. Another important factor affecting the efficiency of bacteria swimming is the concentration of dissolved, at least if bacteria are aerobic. The consumption of oxygen may reduce its concentration in the (narrow channel). It may be negligible or not. But I would like to see some discussion or estimates on it.
4. I would suggest adding the example of the cross-section image from the supplementary file (Figure 2) to the main text, Figure 1.

Reviewer #3 (Remarks to the Author):

This manuscript describes experiments of *E. coli* swimming through microfabricated microchannels. The authors find that *E. coli* exhibits a modest increase in speed as the channels become more confining, up to a wall spacing of about 2.3 microns. Beyond this point, normalized bacterial speed

slows down. The authors show that the likely cause of the increase in speed is due to the bacteria swimming more along the center of the channel as the channel becomes more constricting. Since the center of the channel is the furthest point from the walls, this location is where drag is minimized, allowing the bacteria to swim faster. The authors develop a model that is in fair agreement with their results on the change in speed as a bacterium swims out of a channel. This model assumes that the flagellar motors exert constant torque and uses simulations to estimate drag on the bacteria while they are in the channels. Overall, the results are somewhat interesting; however, I am not convinced that they are substantial enough to warrant publication in Nature Communications and may be better suited for a more specialized journal. Specifically, the main results seem to be mostly confirmation of things like the increase in drag the closer a bacterium swims to a wall and that the flagellar motor is well modeled by assuming constant torque. The one main new observation is the stable swimming down the axis of the channel for narrow channels.

Specific comments

1. I am not sure one can classify flagellar swimming as probably "the simplest form of locomotion in the living world." I would suggest removing this statement from the Abstract.
2. What is the exact geometry and dimensions of the turning sections between the microtunnels? Is it possible that the shape of the device produces any focusing of the bacterial motility? For example, if one made a similar device but with all the tunnels having the same width, would the results for the first tunnel be indistinguishable from the last tunnel? Or, similarly, if the narrowest tunnel came first and the widest last would the results be the same as are presented here?
3. Over what distance was the "free" swimming speed measured? What was the average free swimming speed? How much variation is there in swimming speed over a single cell's trajectory? What is the variation over the population? The same questions on variation in speed apply to speed within the tunnels.
4. I assume that the error bars on the average normalized speed shown in Figure 2a are smaller than the marker size. Is this correct. If it is, the authors should mention it in the caption. If not, the error bars should be included.
5. Are there correlations between cell wobble and mean cell angle? What about for wobble and cell angle as a function of tunnel width? Where are the wobble and cell angle measured? Does a cell that wobbles significantly in the channel also wobble significantly during its "free" swimming period?
6. Is the "stable" axial swimming an artifact of the fact that for narrow channels, a cell that wobbles is effectively fatter than its half-width. Therefore, wobbling gives the cell a larger effective diameter and forces the cell centroid to remain close to the tunnel axis?
7. In the Abstract, the authors claim that swimming bacteria can suggest effective design principles for microrobots. However, since the results presented here are on a single strain of *E. coli*, I don't see any way to try to use the results to guide design of microrobots. In order to do this, would require exploring how different geometries, etc. of bacteria fair in these channels.

Authors reply to Reviewers reports

We would like to sincerely thank all reviewers for the many interesting questions and suggestions. We believe the manuscript is much better now and we really hope it can be published in Nature Communications in its revised form. Here follows a detailed list of replies to all comments raised by the Reviewers.

Reviewer #1

In this work, Vizsnyiczai, Frangipane, Di Leonardo et al. present the use of 3D printed microchannels of decreasing size to study the effect of constriction on the movement of a non-tumbling E. coli strain. The authors show that a channel size of 2.3 μm is optimal for high-speed navigation and the dependency of the swimming speed is correlated to the position of the cell within the channel.

The experimental set-up and the subject is of high relevance for the microrobotic community. However, there are several points that require clarification before publication can be granted. Below, you find a list of such points:

- **The titles is very general and points towards bacterial motility and narrow channels. Here, only a very limited set of conditions is tested. This reduced set clearly limits the validity of the work actually done and/or calls for a change of the title.**

The revised manuscript includes new experiments and theoretical analysis that expand the set of tested conditions (i.e. channels with different cross sections). However we rephrase the title to a less general form that explicitly refers to E. coli, recognizing that swimming through the rotation of a helical flagellar bundle is an important ingredient for the observed phenomenology.

A transition to stable 1D swimming enhances E. coli motility through narrow channels

- **This also points towards some questions regarding the claim of the authors from a biological point of view in the absence of proper controls: The authors only use a mutant E. coli strain. I fully understand the practical aspects associated with this type of bacteria, however, do the wild-type E. coli or the K12 mutant and other species show the same behavior?**

The choice of a smooth swimming mutant strain is not just for practical reasons but it was dictated by the necessity of isolating and highlighting only extrinsic hydrodynamic effects on flagellar propulsion excluding the role of specific intrinsic factors. Wild type strains alternate swimming in straight runs with tumbling events which may affect experiments in different ways:

1. tumbling events are not always easy to spot and may cause spurious large speed fluctuations that are due to intrinsic cell factors and not to external confinement effects
2. smooth swimming cells also ensure a short transit time and reduce the risk of structure clogging due to bacterial collisions

We think that such a purely hydrodynamic phenomenon might be common to other species of flagellated bacteria which calls for future experiments. These experiments are however not straightforward involving the construction of an equivalent smooth swimming strain.

This is now also explained in the text as follows:

*We use a non-tumbling *E. coli* strain so that cells can swim smoothly throughout the channels reducing the chance of occasional collisions and clogging of the channels. Furthermore, tumbling events in wild type cells are not always easy to detect and may cause spurious speed fluctuations which are not related to the effects of external confinement.*

- **Along the same line, working with another species would give information on the parameters used for this type of cell. Is the 2.3 μm a general feature or it is related to the cell size and / or to the flagellar apparatus dimension? What about the channel geometry? Would another morphology (circle cut) have a similar outcome? These questions have been ignored and should be commented if not directly tackled.**

We thank the Reviewer for stimulating further work in this direction. The threshold channel size will be dependent on the specific geometric characteristics of the swimmer (body length and thickness, flagella pitch, thickness, diameter and length). Rather than parametrically exploring all possible values, we have chosen to perform numerical simulations to check the existence of stable axial swimming for swimmers having the geometrical characteristics of an *E. coli* cell. With our great surprise and enjoyment we found not only that axial swimming is stable but that the transition to stable swimming occurs precisely where experimentally observed. We exploited the versatility of simulation to address the point raised by Reviewers 1 and 2 about the robustness of results with the shape of channel cross-section. As now reported in the manuscript the transition occurs both for circular and square channels at the same threshold value for the cross-section area. A new paragraph is added at page 4 with a new related figure appearing in the main text as Fig. 4.

- **The authors only study the paths of the bacteria towards smaller channels. Even for microorganisms, a “fatigue” can be observed. Therefore, the authors have to analyze the track of bacteria starting from smaller channels towards larger channels and check if similar results are obtained. On the same line, the authors analyze the speed of the cells while exiting the channel, they have to provide data for the bacteria entering the channel too and compare the entry speed with the exit speed.**

We thank the Referee for raising the two points above. We also agree that questions of this kind deserve clarification by direct experimental observations. To this end we now report new experimental data that have been collected using a “control” structure made of channels of equal size. No sign of “fatigue” is observed as now discussed in the main text:

As a control experiment we fabricated a second structure composed of equal sized tunnels (width $3 \mu\text{m}$) to check for the presence of any intrinsic cellular factor that could give rise to a systematic speed variation as the cell traverses the entire structure. Results are reported in the inset of Fig. 2a and show no statistically significant differences between group means as determined by one-way ANOVA ($p=0.95$).

- **What is the statistical relevance of this work? This may be more a biologist question as it might be clear for the physicists but the authors limit their work to 100 tracks. Again, I understand it may be difficult to track the bacteria in these conditions, but due to the fact that the authors based all their claims only working on average value of highly diverging data, they should provide a statistical analysis attesting the relevance of the claims. Indeed, the average normalized speed difference reported in the channels varies from about 0.9 to 1.1 for channel sizes varying from 2 to 4 μm whereas it seems the actual normalized speed spans from 0.6 to 1.4.**

We now discuss statistical relevance of speed data in the following paragraph:
Despite the large speed differences among cells, the standard error of normalized mean speeds in Fig. 2a is always within the symbols size. The observed initial speed increase is statistically significant with a p-value for the comparison between tunnels $2.3 \mu\text{m}$ (orange) and $3.9 \mu\text{m}$ (blue) given by $p=10^{-8}$ (Mann-Whitney).

Reviewer #2:

The submitted manuscript is focused on the hydrodynamics of bacterial swimming in relatively narrow channels of different width. The authors report the increase in the swimming velocity of an individual bacterium if the channel width is around 2-3 micron. This increase is associated with the transition to stable swimming straight along the center line of the channel. I believe that the paper provides a new and important experimental input for understanding the bacteria hydrodynamics in microchannels and may be considered for publication in Nature Communications after the author clarified several points. Main points to address. The results on bacterial swimming velocity vs channel width are presented in a clear and well-organized manner. The authors emphasize some apparent contradictions with previous studies [14] and resolve it by pointing to a difference in positions of the

bacterium inside the channel. While this is correct, I believe that there is an opportunity to increase the depth of analysis by presenting additional data.

- **It would be nice to track the flagella rotation rate vs channel width. It is interesting to see how the frequency changes with the width. If the frequency does not peak at ~2.2 um width then the bacterium is swimming more "efficiently", traveling farther per revolution of flagella due to change in the shape of microscopic hydrodynamic flows. The rotation rate for the same bacterium is at a nearly constant ratio with the bacteria wobbling frequency. Since the amplitude of wobbling was tracked I assume that this data on frequency should be relatively easy to extract from already collected experimental observations. Alternatively, the author may perform another series of experiments with fluorescently marked flagella. It will provide more accurate and detailed data.**

The possibility of directly observing fluorescently labelled flagella inside channels could clearly add interesting details to support our claims. That's why we tried to repeat experiments with a smooth swimming strain (HCB1737 Δ CheY) carrying a mutated flagellin gene that can be selectively stained. Unfortunately, for reasons that we failed to understand, these cells end up getting stuck in the channels that are eventually clogged as shown in the following figure. In the figure a superposition of epifluorescence and bright-field microscopy is used to visualize both the flagella and the tunnels.

We also would like to remark that tracking the flagella rotation requires a framerate of at least 200 fps. Considering the high excitation intensity required to visualize the fluorescence at that framerate and the long time interval required for a bacterium to go through the entire structure, it is hard task to observe the flagellum in all the tunnels without bleaching entirely the fluorescence.

Although we are working to solve this issue in the next future, the main claims of the paper (transition to stable axial swimming + enhanced thrust) are still well supported by the reported data. As the Reviewer correctly suggests the wobbling frequencies can be extracted and we now report them in a new Figure appearing in the Supplementary Material. Before discussing this data we want to remark that it is not always true that “the rotation rate for the same bacterium is at a nearly constant ratio with the bacteria wobbling frequency”. This might be the case when changing a fluid property (such as viscosity in Newtonian fluids) that affects the rotational drag coefficient of body and bundle by a common factor. However confinement can affect the drag coefficients of body and bundle very differently. As an example, as reported in the manuscript, we estimate that while the translational drag on the bundle

increases by a factor 1.54 the drag increase on the body is 3.85. Although not directly connected to flagellar rotation frequency we now expanded the Supplementary Information with a new section on wobbling dynamics featuring two new figures. However we think the Reviewer's questions on efficiency address an issue that deserves to be discussed in the manuscript. Stimulated by the Reviewers comments we actually realized that there is something interesting to say about efficiency even in the absence of data on flagellar frequency:

At this point it comes natural to wonder if swimming efficiency is also enhanced by confinement. There are mainly two ways of defining the self-propulsion efficiency of a swimmer. The first one is by the ratio of swimming speed U over the bundle rotational frequency $\omega/2\pi$. This ratio has the dimensions of a length and represents the distance traveled by the cell in a full rotation period of the bundle. Assuming a constant torque, the energy supplied by a flagellar motor in a full rotation cycle is constant so that this definition corresponds to what we generally use for transport vehicles, i.e. the ratio of distance traveled per unit of fuel consumed. With a little bit of manipulation this efficiency can be expressed as $\epsilon_1 = 2\pi U / \omega = 2\pi b U / \tau \propto b U$ where we have used $\tau \approx b \omega$. Alternatively, swimming efficiency is often defined as the ratio between the power required to drag a dead cell body at a speed U and the power $\tau \omega$ supplied by the flagellar motors to self-propel the cell at the same speed $\epsilon_2 = (A+a)U^2 / \tau \omega = c U / \tau \propto c U$ where we have used the approximation $U \approx c \tau / b(A+a)$ in (ref{Eq:speed}). It is interesting to note that both efficiencies are proportional to the product of cell speed by a drag coefficient in the bundle resistance matrix. If the drag coefficients increase under confinement, it can be concluded that a higher swimming speed also corresponds to a higher swimming efficiency. The situation is not so straightforward in the last two channels where the speed falls below one but the corresponding efficiencies could still be greater than outside if the coefficients c and b are sufficiently larger than in "free" cells. In the case of the last tunnel we have reliable estimates for the ratios c^{\prime}/c^0 and b^{\prime}/b^0 obtained from the fit of data in Fig.\ref{fig:f5}. Substituting those values we can obtain the efficiency ratio between the last tunnel and the free case. We find that the first efficiency decreases in the tightest tunnel $\epsilon_1^{\prime}/\epsilon_1^0 = (b^{\prime}/b^0)(U^{\prime}/U^0) = 0.68$. The second efficiency is remarkably still the same as outside $\epsilon_2^{\prime}/\epsilon_2^0 = (c^{\prime}/c^0)(U^{\prime}/U^0) = 1.95 \times 0.5 = 1$ indicating that flagella shape seems to be optimized for maximum efficiency even under strong confinement conditions \cite{Liu2014}.

- **On the downside, I found the discussion of the thrust increase in a narrow channel by analysis of speed variation near the channel exist a little weak and speculative. First of all, I suggest to define the thrust as "the force that is required to keep the cell at rest " before the analysis, not after. Otherwise, the thrust may be assumed to be equal to the drag in magnitude and opposite in direction. Then it is not clear why it's not changing between points I and II.**

We agree on the need of defining thrust clearly and earlier. This is also most likely the cause for the Reviewer's misunderstanding of our claim: between points I and II, the thrust f_t (a bundle only property) does not change while the speed ($U=f_t/(A+a)$) goes up as the drag A on the cell body decreases upon exiting the tunnel. Text now reads as:

We define the thrust as the force transmitted by the rotating bundle to the cell body when the cell body is kept fixed by an external force ($U=0$). From Eq. 3, this force is $f_t=-c\tau/b$. This definition of thrust has the advantage of being a property of the bundle alone with no reference to the actual drag on the load (cell body).

By imposing force free ($F+f=0$) and torque free ($T+\tau=0$) conditions we can solve for the cell speed: $U=f_t/(a+A)$

- **Second, the author did not provide expressions for $a(x)$, $b(x)$ and $c(x)$ calling them "similar". That confused me. I can assume these expressions should include flagella length as a parameter, which should also be fitted.**

We now define explicitly the expressions for $a(x)$, $b(x)$ and $c(x)$ that, as correctly pointed out by the Reviewer, contain the bundle length. In the previous manuscript version the flagellar bundle length ℓ was estimated from data shown in Fig.5 as the distance travelled by cells to go from point II to point III. In the revised manuscript ℓ is an additional fitting parameters as explained in the text:

The parameters A^0 , a^0 , b^0 , A^{\prime} , a^{\prime} , b^{\prime} are obtained from numerical calculations (see Methods) over a range of ℓ values while c^0 and c^{\prime} are left as free fitting parameters. The best fit curve is plotted as a solid line in Fig.\ref{fig:f5} and provides a very good representation of data points. We find a value for $\ell=6.7\pm 0.3\ \mu\text{m}$ that is compatible with literature data.

- **Finally, the assumption that the torque applied to flagella is constant is weak. The torque is indeed nearly a constant (with some limitations) under a relatively heavy load. A freely swimming bacterium does not utilize the maximum torque, therefore it is not constant under any load. (Moreover, it may have some memory effect and does not change instantaneously but it's the next degree of complexity). I would like to hear the authors comment on that.**

As far as we understand from [1], at room temperature, the flagellar motor provides a constant torque when the rotation frequency of the flagellum f is smaller than a corner frequency of approximately 200 Hz. We estimate that rotation speed as $\omega=U(A+a)/c$ therefore we expect that for our faster cell ($U=40\ \mu\text{m/s}$) is $f=186\ \text{Hz}$ while for our average cell ($U=27$) we have $f=126\ \text{Hz}$.

However we decided to look for a further experimental check of the constant torque assumption. In the Supplemental Material we added a new figure plotting the time evolution of the cell body axis angle for several cells coming out of the last tunnel. The rotation speed of the cell body is $\Omega=T/B$. When the cell body is out of the

tunnel $B=B_0$ therefore Ω is constant only if the torque provided by the motors is also constant. Fig. 9 of the Supplementary Material shows no visible change of the wobbling frequency soon after the cell body is out of the tunnel. As the bundle progressively comes out of the tunnel, its rotational drag will decrease and its rotation frequency will change but the provided torque T remains constant.

[1] Chen, Xiaobing, and Howard C. Berg. "Torque-speed relationship of the flagellar rotary motor of *Escherichia coli*." *Biophysical journal* 78.2 (2000): 1036-1041.

Other questions and suggestions:

- **The vertical profile of the channel is not rectangular but has rounded corners. What is a typical radius? When the size of a channel is decreasing, this may become important and affect the localization of a bacterium inside the channel. How important is the transition from rectangular to circular confinement?**

This issue had already been raised by the first Reviewer. Following their suggestion we have investigated this point further. Both the main text and the Supplementary Material have been significantly expanded to include new data analysis and simulations demonstrating that the transition to a stable axial swimming occurs for both circular and square channels (see reply to the third point of Reviewer 1 for more details). In light of these new results, we are quite confident that the details of the shape of the channel do not play a crucial role.

- **How do the authors define the position of the wall? Was it done by processing the images similar to Figure 2d?**

Yes. The exact procedure is now described in detail in the Methods section:

The position of the wall appearing in Figs. 2c and 3b can be accurately determined by aligning the cells trajectories in a coordinate system where the center of the smallest tunnel is located at $x=0$. Small drifts of the sample are corrected by setting to zero the mean position of each cell in the smallest tunnel. The wall positions in this coordinate system are then calculated by considering the spacing ($15 \mu\text{m}$) of the microtunnels and their widths which are extracted from SEM images and two-photon microscopy as described in the Supplementary Information and reported in Fig. 2d.

- **Another important factor affecting the efficiency of bacteria swimming is the concentration of dissolved, at least if bacteria are aerobic. The consumption of oxygen may reduce its concentration in the (narrow channel). It may be negligible or not. But I would like to see some discussion or estimates on it.**

That is an interesting effect to consider and whose size is also quite easy to estimate. Oxygen consumption is now discussed in a footnote of the main text:

For example, one might think that oxygen can be depleted from the surrounding environment when the cell swims in a narrow tunnel. However, a simple estimate already shows that this effect is negligible. Typical oxygen consumption rates in *E. coli* is $Q \approx 20 \text{ amol/min/cell}$ [Schwarz2016escherichia]. Assuming steady state diffusion this consumption rate will be equal to the oxygen flow through the tunnel cross-section. The resulting relative concentration variations will be then $\Delta C/C \approx Q l / (C D s^2) = 10^{-3}$ where we used $D \approx 2 \cdot 10^{-5} \text{ cm}^2/\text{s}$ for oxygen diffusion coefficient in water [han1996temperature], $C = 1 \text{ amol}/\mu\text{m}^3$ for oxygen concentration in water [carpenter1966new] while l and s^2 are typical length and cross-section area of our tunnels.

[1] Schwarz-Linek, Jana, et al. "Escherichia coli as a model active colloid: A practical introduction." *Colloids and Surfaces B: Biointerfaces* 137 (2016): 2-16.

[2] Han, Ping, and David M. Bartels. "Temperature dependence of oxygen diffusion in H₂O and D₂O." *The Journal of physical chemistry* 100.13 (1996): 5597-5602.

[3] J.H. Carpenter "New measurement of oxygen solubility in pure and natural water" *Limnol. Oceanogr.*, 11 (1966), pp. 264-277

- I would suggest adding the example of the cross-section image from the supplementary file (Figure 2) to the main text, Figure 1.

We thank the Reviewer for this suggestion, we added cross-section images in Fig.2 which clearly improves data presentation.

Reviewer #3

This manuscript describes experiments of *E. coli* swimming through microfabricated microchannels. The authors find that *E. coli* exhibits a modest increase in speed as the channels become more confining, up to a wall spacing of about 2.3 microns. Beyond this point, normalized bacterial speed slows down. The authors show that the likely cause of the increase in speed is due to the bacteria swimming more along the center of the channel as the channel becomes more constricting. Since the center of the channel is the furthest point from the walls, this location is where drag is minimized, allowing the bacteria to swim faster. The authors develop a model that is in fair agreement with their results on the change in speed as a bacterium swims out of a channel. This model assumes that the flagellar motors exert constant torque and uses simulations to estimate drag on the bacteria while they are in the channels. Overall, the results are somewhat interesting; however, I am not convinced that they are substantial enough to warrant publication in *Nature Communications* and may be better suited for a more specialized journal. Specifically, the main results seem to be mostly confirmation of things like the increase in drag the closer a bacterium swims to a wall and that the flagellar motor is well modeled by assuming constant torque. The one main new observation is the stable swimming down the axis of the channel for narrow channels.

Specific comments

1. I am not sure one can classify flagellar swimming as probably "the simplest form of locomotion in the living world." I would suggest removing this statement from the Abstract.

We now use the weaker form:

..., *bacteria employ one of the simplest forms of locomotion in the living world.*

2. What is the exact geometry and dimensions of the turning sections between the microtunnels? Is it possible that the shape of the device produces any focusing of the bacterial motility? For example, if one made a similar device but with all the tunnels having the same width, would the results for the first tunnel be indistinguishable from the last tunnel? Or, similarly, if the narrowest tunnel came first and the widest last would the results be the same as are presented here?

This is a crucial point that had been raised already by Reviewer 1. We followed the Reviewer suggestion and studied bacteria swimming in a "control structure" having all tunnels of the same width. Results are now discussed in the main manuscript and also reported as an inset in Fig.2a. Based on these new findings we can safely say that there is no statistically significant difference between different channels having the same width.

3. Over what distance was the "free" swimming speed measured? What was the average free swimming speed? How much variation is there in swimming speed over a single cells trajectory? What is the variation over the population? The same questions on variation in speed apply to speed within the tunnels.

All this information is now included in the manuscript:

The variability of cells' speed in each tunnel is 15% (relative standard deviation). After the cells exit the last tunnel, they are further tracked along a path of about 50 μm to extract their "free" speed. The mean "free" speed is 28.8 $\mu\text{m/s}$ with a standard deviation of 3.8 $\mu\text{m/s}$.

When averaged over a long enough trajectory, the mean speed is a well defined quantity. Conversely, the instantaneous speed variability is framerate dependent as it is strongly affected by the tracking error as well as Brownian fluctuations. Therefore we do not include instantaneous speed variability in the text.

4. I assume that the error bars on the average normalized speed shown in Figure 2a are smaller than the marker size. Is this correct? If it is, the authors should mention it in the caption. If not, the error bars should be included.

We now discuss statistical relevance of speed data in the following paragraph:

Despite the large speed differences among cells, the standard error of normalized mean speeds in Fig. 2a is always within the symbols size. The observed initial speed increase is statistically significant with a p-value for the comparison between tunnels 2.3 μm (orange) and 3.9 μm (blue) given by $p=10^{-8}$ (Mann-Whitney).

5. Are there correlations between cell wobble and mean cell angle? What about for wobble and cell angle as a function of tunnel width? Where are the wobble and cell angle measured? Does a cell that wobbles significantly in the channel also wobble significantly during its "free" swimming period?

All requested information is now available in Supplementary Note 4. Wobble and cell angle are measured in the largest tunnel as now explicitly stated in the manuscript and Fig.3 caption.

6. Is the "stable" axial swimming an artifact of the fact that for narrow channels, a cell that wobbles is effectively fatter than its half-width. Therefore, wobbling gives the cell a larger effective diameter and forces the cell centroid to remain close to the tunnel axis?

The transition to axial swimming happens sharply for all cells once the tunnel size is below 2.3 microns. In that situation, there is still a consistent gap between the cell and the tunnel wall (see Fig.2b). A further experimental evidence in favour of an explanation based on hydrodynamics is shown in Fig.3b where the cell-wall gap is plotted as a function of the tunnel size. For weakly wobbling cells, the gap increases when the tunnel size decreases from 4 μm to 2.3 μm .

7. In the Abstract, the authors claim that swimming bacteria can suggest effective design principles for microrobots. However, since the results presented here are on a single strain of E. coli, I don't see any way to try to use the results to guide design of microrobots. In order to do this, would require exploring how different geometries, etc. of bacteria fair in these channels.

While it is true that the geometrical parameters of flagella can be different amongst different species of bacteria, the propulsion mechanism is a general property of the helical shape. In line with this, one class of synthetic microswimmer robots were developed to harness the same propulsion hydrodynamics as bacteria with helical flagella [1,2,3].

Our results provide new insight into effects that have to be taken into account in the operation of such microrobots, showing that they can function with an increased propulsion efficiency in strongly confined environments, and they may self-align on the channel axis to swim swiftly through narrow constrictions such as blood vessels.

We have slightly modified the last sentence of the abstract hoping to state this more clearly:

Our results challenge current theoretical predictions and may also suggest effective design principles for microrobots by showing that motility based on helical propellers provides a highly robust swimming strategy for exploring narrow spaces.

[1] Qiu, Famin, et al. "Magnetic helical microswimmers functionalized with lipoplexes for targeted gene delivery." *Advanced Functional Materials* 25.11 (2015): 1666-1671.

[2] Yan, Xiaohui, et al. "Multifunctional biohybrid magnetite microrobots for imaging-guided therapy." *Science Robotics* 2.12 (2017).

[3] Schuerle, S., et al. "Synthetic and living micropropellers for convection-enhanced nanoparticle transport." *Science advances* 5.4 (2019): eaav4803.

REVIEWERS' COMMENTS:

Reviewer #1 (Remarks to the Author):

I thank the authors for critically addressing the point raised in my initial review. I fully understand that not all possible geometries and bacterial species can be tested. If I doubt that the smooth swimmers are biologically so relevant, the case is certainly interesting for potential microrobotic applications validating the work presented here. There is only one point that has apparently remained unexplored: the case of bacteria starting from the smaller sized channels. I am unsure that the control with channels of equal size answers the question of the focusing of swimming inside the channel due to the decreasing channel size.

Reviewer #2 (Remarks to the Author):

The authors addressed my concerns. The edits and new materials (both figured and discussions) significantly improved the overall presentation in the updated version of the manuscript. I must note there are still some minor deficiencies related to not fully supported claims. For example, I don't understand the new discussion in SI on a constant torque. Why do authors assume that the rotation drag does not change with the channel width or during the exit? However, I agree that the detailed investigation of the constant torque question is the main topic of this paper and is a separate problem. While the paper has a couple of weak places, I believe that the main effect is new, interesting, clearly presented. This work may be interesting for researchers focused on microfluidics, active matter and or even bacterial infections. Therefore, I think that this paper may be accepted for publication.

Reviewer #3 (Remarks to the Author):

The authors have adequately addressed my concerns.

Authors reply to Reviewers reports

We thank the Referees to acknowledge that we addressed all the points raised in their first report. Here we reply to the two comments that have been made in the last report.

Reviewer 1

There is only one point that has apparently remained unexplored: the case of bacteria starting from the smaller sized channels. I am unsure that the control with channels of equal size answers the question of the focusing of swimming inside the channel due to the decreasing channel size.

In the previous Reviewers report, Reviewer 3 was pointing out this issue of focusing and the consequent need of performing control experiments saying:

“Is it possible that the shape of the device produces any focusing of the bacterial motility? For example, if one made a similar device but with all the tunnels having the same width, would the results for the first tunnel be indistinguishable from the last tunnel?”

We thought this was a relevant point and addressed it with a new control experiment that was exactly what was asked by the Reviewer. We believe that, given the strong correlation between the speed and the distance from the wall evidenced in Fig.2b the finding of a constant speeds in equal sized channels provides a convincing evidence that the onset of axial swimming is not due to a hydrodynamic focusing. In addition to this important control experiment, the previously revised manuscript also includes a new numerical model that provides a solid theoretical background for our claims.

Reviewer 2

I don't understand the new discussion in SI on a constant torque. Why do authors assume that the rotation drag does not change with the channel width or during the exit?

Indeed the rotational drag of both the cell body and flagellar bundle changes both with channel width and as the body or bundle move progressively out of the channel. What we assume is that the rotational drag on the cell body does not change *after it is completely out of the tunnel*. This was probably not stressed well enough in the previous version and we have now expanded the SI main text with a more carefully description of Fig.9:

Fig.9 plots the body angle as function of time of several cells coming out from the tightest channel. Time origin is shifted so that for $t < 0$ (shaded area) the cell body is inside or partially inside the tunnel. Conversely, for $t > 0$, the cell body is completely out of the tunnel while the bundle is progressively coming out. The transition from confined to free cell body is

accompanied by an increase of the wobbling amplitude. Each orange line in Fig. 9 fits with constant frequency oscillations the cell angle dynamics soon after the cell body is completely out. When the cell body is completely out of the tunnel, the rotational drag of the body B_0 is constant therefore a constant frequency corresponds to a constant torque ($\Omega = T/B_0$).